# A Picrocrocin-Enriched Fraction from a Saffron Extract Affects Lipid Homeostasis in HepG2 Cells through a Non-Statin-like Mode

**DOI:** 10.3390/ijms24043060

**Published:** 2023-02-04

**Authors:** Luca Frattaruolo, Federica Marra, Graziantonio Lauria, Carlo Siciliano, Rosita Curcio, Luigina Muto, Matteo Brindisi, Donatella Aiello, Anna Napoli, Giuseppe Fiermonte, Anna Rita Cappello, Marco Fiorillo, Amer Ahmed, Vincenza Dolce

**Affiliations:** 1Department of Pharmacy, Health and Nutritional Sciences, University of Calabria, 87036 Arcavacata di Rende, Italy; 2Department of Chemistry and Chemical Technologies, University of Calabria, 87036 Arcavacata di Rende, Italy; 3Department of Biosciences, Biotechnologies and Environment, University of Bari, 70125 Bari, Italy

**Keywords:** picrocrocin, saffron extract, low-density lipoprotein receptor (LDLR), lipid metabolism, hypolipidemic activity, HepG2 cells

## Abstract

Dyslipidemia is a lipid metabolism disorder associated with the loss of the physiological homeostasis that ensures safe levels of lipids in the organism. This metabolic disorder can trigger pathological conditions such as atherosclerosis and cardiovascular diseases. In this regard, statins currently represent the main pharmacological therapy, but their contraindications and side effects limit their use. This is stimulating the search for new therapeutic strategies. In this work, we investigated in HepG2 cells the hypolipidemic potential of a picrocrocin-enriched fraction, analyzed by high-resolution ^1^H NMR and obtained from a saffron extract, the stigmas of *Crocus sativus* L., a precious spice that has already displayed interesting biological properties. Spectrophotometric assays, as well as expression level of the main enzymes involved in lipid metabolism, have highlighted the interesting hypolipidemic effects of this natural compound; they seem to be exerted through a non-statin-like mechanism. Overall, this work provides new insights into the metabolic effects of picrocrocin, thus confirming the biological potential of saffron and paving the way for in vivo studies that could validate this spice or its phytocomplexes as useful adjuvants in balancing blood lipid homeostasis.

## 1. Introduction

Dyslipidemia is a lipid metabolism disorder depending on multiple factors, such as genetic predisposition, metabolic capacity, and dietary intake [1]. Dyslipidemia is characterized by the presence of one or more of the following phenotypes: elevated serum concentrations of low-density lipoproteins (LDL), triglycerides (TG) and total cholesterol, or low concentrations of high-density lipoproteins (HDL). The main manifestation of dyslipidemia is atherosclerosis, which can lead to cardiovascular diseases, which are leading causes of death in industrialized countries. These pathologies include coronary heart diseases (angina pectoris and heart attack), cerebrovascular (stroke) and peripheral vascular diseases [2]. Atherosclerosis is a degenerative pathological condition affecting medium and large caliber arteries and causing an accumulation of fat and white blood cells in the inner wall of the blood vessels, in which atherosclerotic plaques are formed [3]. The most predisposed subjects are those with one or more risk factors, such as diabetes, hyperlipidemia, hypertension and obesity; in addition, smoking, alcohol, sedentary life, incorrect diet, age and gender can contribute to the onset of these diseases.

The therapeutic approach to treating different metabolic diseases such as cancer, inflammation and dyslipidemia primarily involves changes in lifestyle and diet [4,5,6,7,8,9]; if this is not enough, the use of drugs is recommended. One of the main pharmacological strategies used in clinical practice is to target β-hydroxy-β-methylglutaryl-CoA reductase (*HMGR*), an enzyme that plays a crucial role in cholesterol biosynthesis regulation [10]. Statins are hypocholesterolemic drugs acting as competitive inhibitors of *HMGR*, thus lowering LDL levels. Although statins are able to reduce cardiovascular risk in the range of 15 to 37%, a substantial residual risk remains due to insufficient LDL lowering, high TG and low HDL levels [11,12,13,14]. Furthermore, statin administration is not always possible; indeed, more than 40% of patients experience important side effects that make this therapeutic approach unsuitable [15,16,17]. These side effects include altered liver function, muscle pain and gastrointestinal disorders; moreover, statin administration should be avoided in alcoholics, pregnant and breastfeeding women, as well as in children and patients with liver dysfunction. Furthermore, these drugs should not be combined with fibrates [13,14], as they can lead to myopathy, rhabdomyolysis and renal failure. Their administration is also not recommended in patients with type 2 diabetes mellitus and metabolic syndrome [18], as although they promote a significant reduction in LDL levels, they are not effective in reducing the onset of cardiovascular diseases. In this regard, it has been observed that over 50–60% of patients treated with statins experience cardiovascular pathologies within 5 years of treatment [19].

Due to the different contraindications and side effects of statins, research is now very active in identifying new molecules that are endowed with reduced toxicity and are more accessible to patients regardless of their risk factors. In this regard, research attention has shifted towards plant extracts and natural products, to which are attributed the ability to modulate lipid metabolism, and which are usually well tolerated, having low or no toxicity [7,20].

Saffron is a precious spice obtained from the stigmas of *Crocus sativus* L., a plant belonging to the Iridaceae family and which mainly grows in Asian countries, in particular in Iran, Iraq and India, as well as in some Mediterranean countries such as Greece, Spain and Morocco [21]. This spice is characterized by the presence of three main metabolites: crocin, responsible for the yellow color of the stigmas; picrocrocin, which determines the bitter taste of saffron; and safranal, a volatile terpene aldehyde responsible for the characteristic smell and aroma of this spice. Crocin and picrocrocin are the major compounds found in saffron, and picrocrocin represents the biosynthetic precursor of safranal. Other natural compounds found in saffron are anthocyanins, flavonoids, vitamins (riboflavin and thiamine), amino acids, proteins, starch and minerals [22]. Saffron has several biological properties and is considered a potential therapeutic drug. For this reason, it is widely used as an anti-catarrhal, anti-spasmodic, nerve sedative, diaphoretic, carminative and expectorant [23]. The antioxidant effects of saffron are attributed to the presence of crocin and safranal, which exhibit significant radical scavenging activity [24,25,26,27]. Starting from its antioxidant action, other effects have emerged, such as anti-cancer, anti-toxic [28,29,30], anti-nociceptive [31,32] and anti-inflammatory ones [32,33,34]. Saffron promotes a decrease in blood glucose levels and an increase in insulin secretion by pancreatic β-cells; therefore, it appears to be endowed with anti-diabetic and hypoglycemic effects [35,36,37]. In addition, crocin and safranal are believed to exert antidepressant effects [38,39], since crocin may inhibit dopamine and norepinephrine uptake, while safranal can affect the serotonergic system.

Furthermore, recent studies have highlighted the hypolipidemic [40] and anti-atherosclerotic [41,42] activity of crocin. In detail, it is able to selectively inhibit pancreatic lipase, acting as a competitive inhibitor. This leads to a reduction in the absorption of fats and cholesterol, thus lowering blood levels of TG, total cholesterol, LDL and very low-density lipoproteins (VLDL).

Since there are currently no studies on the effects of picrocrocin on lipid metabolism, in this work, we aimed to investigate the activity of a picrocrocin-enriched fraction (PEF) from saffron, without resorting to chemical-physical manipulations, in order to evaluate the potential additional hypolipidemic effect that this natural product can exert in saffron phytocomplexes. In order to evaluate and characterize the biological activity of this natural compound, the effects of PEF were studied in the HepG2 cell line, i.e., hepatocytes widely used as an in vitro model for the study of cell metabolism.

## 2. Results and Discussion

### 2.1. High-Resolution ^1^H NMR Spectrum of PEF

In order to prepare this type of extract, a well-known, rapid and straightforward extraction methodology was applied to the commercially available matrix, which is also useful in the case of other different complex plant matrices [43,44,45]. The high-resolution ^1^H NMR spectrum of PEF was characterized by a high complexity. A series of well-distinguishable resonance signals were detectable; however, many signals strongly overlapped or appeared as shoulders of the most intense peak, centered at 3.41 ppm, which was generated by the methyl protons of methanol used for separation. The peak at 2.51 ppm was generated by the residual resonance of unlabeled DMSO-d6, which was employed to calibrate the spectrum. The latter could be divided into four regions, each one containing signals which could be referenced to the different chemical components of the extract. The first region between 0.50 and 2.70 ppm displayed signals attributable to fatty acid chains (0.86, 1.46, 1.74 ppm), and a pair of singlets attributable to the protons of 7 and 8 methyl groups in the structure of picrocrocin, at 1.16 and 1.19 ppm, respectively. The singlet that partially overlapped at 1.98 ppm was attributed to the methyl group at the 9 position of picrocrocin, while the series of the clearly visible multiplets between 1.50 and 1.80 and 2.10 and 2.60 ppm confirmed the resonances of the 5-CH_2_ and 3-CH_2_ protons, respectively, in the picrocrocin structure. The singlet at 2.10 was due to the presence of acetic acid in the sample. A second spectral region between 2.80 and 5.50 ppm displayed a series of peaks that strongly overlap. This was the spectral window, containing all signals mainly due to the resonances of OH functions and CH protons of the β-glucose residue of picrocrocin. The multiplet at 4.01 ppm was the signal correlated to the resonance of 4-CH in picrocrocin. The primary OH of β-glucose was very well-detectable as a triplet centered at 4.20 ppm. The partially overlapped peak at 5.19 ppm was attributed to the 5′ anomeric CH of the β-glucose ring. The absence of safranal, the unsaturated aglycone of picrocrocin, was reportedly undetectable, at least under the limits imposed by the spectral technique. In fact, the narrow chemical shift interval between 5.50 and 6.30 did not feature signals attributable to the protons of the endocyclic double bond. Traces of crocins could not be excluded on the basis of the signals characterizing the fourth spectral window between 6.35 and 7.55 ppm, and were attributable to the protons of the conjugated olefinic chains typically featured in crocins. The intense and well defined singlet at 10.05 ppm proved a likely high amount of picrocrocin in the sample under investigation (Figure 1).

Detection and signal attributions were supported by the literature [46,47] in order to confirm the structures of picrocrocin, safranal, and the other principal methanol soluble saffron metabolites.

The obtained extract showed a quali/quantitative NMR profile (with the relative quantification based on the value of the intensities of the resonance peaks selected for the attribution of the known structures of the compounds of interest), of which the plot contour and values of the signal integrals are close to that illustrated in previous works [43,44,45], thus indicating a picrocrocin content of no less than 90%.

### 2.2. Picrocrocin Inhibits the Activity of the Purified Human Catalytic Fragment (cf-HMGR)

Since *HMGR* is a key enzyme in the regulation of lipid metabolism, and considering that it currently represents the main pharmacological target for the treatment of hypercholesterolaemia and hyperlipidaemia, we decided to evaluate the ability of picrocrocin to inhibit *HMGR* activity. In particular, the inhibitory potential of PEF was assessed on the purified human catalytic fragment (*cf-HMGR*), which was validated as a useful tool for rapid screening of the cholesterol-lowering potential of drug candidates in our previous studies [48]. The results of our spectrophotometric assay showed a statistically significant, albeit modest, inhibitory activity elicited by the PEF from our saffron extract (Figure 2), which was capable of determining 50% inhibition when used at a concentration of 300 µg/mL. The pravastatin concentration used as a control for the *cf-HMGR* activity assay was chosen based on our previous work [48].

The validity of the test was confirmed using pravastatin, a known *HMGR* inhibitor able to reduce enzyme activity in a dose-dependent manner, as a control, halving it when used at a concentration of 100 nM. Based on the modest inhibitory activity found for PEF, and in order to better characterize the pharmacological potential of this natural product, we decided to investigate its effects on lipid metabolism in a more complex model.

### 2.3. Picrocrocin Affects Lipid Homeostasis in HepG2 Cells with a Non-Statin-like Mode of Action

Given the results obtained on the purified *cf-HMGR* enzyme, we decided to evaluate the effects of PEF from our saffron extract on the lipid metabolism of the HepG2 cell line, an in vitro model of hepatocytes widely validated for preclinical studies on cell metabolism. Firstly, a viability test was performed to reveal any toxic effects of the extract on this cell line. As shown in Figure 3A, the obtained results showed that no cytotoxic effect was induced by the treatment of HepG2 cells for 72 h with PEF, up to a concentration of 300 μg/mL.

The ability of PEF to regulate cellular lipid homeostasis was initially evaluated by monitoring transcription levels of the main proteins implicated in the regulation of lipid metabolism, i.e., SREBP1 and SREBP2, which are involved in the regulation of the biosynthesis of triglycerides [49,50,51] and cholesterol [52,53], respectively. HepG2 cells were treated with increasing concentrations of PEF, ranging from 1 to 300 μg/mL. As shown in Figure 3B, our qPCR analysis after 24 h of treatment revealed the ability of PEF to significantly increase *SREBP1* and *SREBP2* transcription levels at the highest concentration tested. This experiment allowed us to choose the optimal treatment condition for further investigations.

Based on these preliminary results, we treated cells using 300 μg/mL PEF, and monitored over time the transcription levels of fatty acid synthase (*FASN*) and glycerol phosphate acyltransferase (*GPAT*), both of which are regulated by *SREBP1*, as well as those of *HMGR* and *LDL receptor* (*LDLR*), both of which are regulated by *SREBP2*. For HepG2 treatment, pravastatin was used as a control, and its concentration was chosen based on previous works [54,55,56]. The obtained results (shown in Figure 4A) revealed an unexpected regulation of the expression of these genes. After 24 h of treatment, the increase in *SREBP2* expression levels was associated, as expected [53], with a marked increase in LDLR transcription levels, whereas surprisingly, a significant reduction in *HMGR* mRNA levels was observed. A similar occurrence was observed for the expression of the genes regulated by *SREBP1*. In particular, the increase in *SREBP1* mRNA levels was found not to be associated with an increase in the transcription levels of the *GPAT* and *FASN* genes.

Prolonging the treatment for 48 h, a reduction in *SREBP1* and *SREBP2* transcription levels was observed, together with a decrease in the transcript levels of their target genes.

Overall, these results show an intense regulation of lipid metabolism by PEF, but with molecular mechanisms that are very different from those used by HMGR inhibitors such as statins. Picrocrocin seems to reduce the synthesis of TG and cholesterol in a time-dependent manner, and this effect is associated with an increase in the *LDLR* expression level, which is responsible for the re-uptake of LDL in the liver under physiological conditions. Conversely, pravastatin inhibits HMGR activity, but is also able to induce an increase in the expression level of the enzyme itself (Figure 4B) through a negative feedback mechanism. This event is experimentally associated with an increase in the expression levels of the enzymes responsible for fatty acids and triglyceride synthesis [57,58].

The immunoblotting analysis of the aforementioned proteins was carried out in order to better understand the regulation of lipid metabolism induced by picrocrocin from our saffron extract. From the obtained results (shown in Figure 5) it can be seen that, although at 24 h, an increase in *SREBP1* expression levels is evident, in our experimental conditions, this protein does not undergo proteolytic activation and therefore remains in an inactive form [59]. This phenomenon allows us to clarify the lack of correlation between increased transcript levels of *SREBP1* and those of its target genes. Conversely, the increase in *SREBP2* expression corresponds to an increase in its cleaved and active form, which is responsible for the increase in the *LDLR* expression level. In particular, after 24 h of incubation, qPCR analysis detected an increase, followed by a dramatic decrease (after 48 h of incubation) in *SREBP2* expression (Figure 4A). Consolidated literature data highlighted that the SREBP-2 protein exerts sterol regulation through cleavage of the membrane-bound precursor protein to release the active form into the nucleus [59]. In this regard, unlike Western blot analysis, qPCR analysis cannot discriminate between the SREBP2 precursor and its cleaved mature form. Based on the qPCR data, the related immunoblot highlighted a marked band due to proteolytic activation of the SREBP-2 precursor (24 h) that dramatically decreased at 48 h (Figure 5A); therefore, the observed increase in the *SREBP2* mRNA level (24 h) followed by its decrease (48 h) could be explained by the level of the SREBP-2 protein mature form.

## 3. Materials and Methods

### 3.1. Chemicals

Solvents (CH_3_CN, and H_2_O, HPLC grade) were purchased from Sigma-Aldrich Fluka (Milan, Italy). Samples of saffron spice were directly obtained from producers, with a guarantee of their origin and freedom from fraud. Dried *Crocus sativus* L. stigmas were obtained from the Cooperative of Saffron (Krokos Kozanis, Greece).

### 3.2. NMR Spectroscopy. Experimental Details

The extract was prepared using a well-known extraction methodology applied to the commercially available matrix, which is widely accepted by the scientific community and is useful for other different complex plant matrices [43,44,45].

Three replicates of PEF were prepared for high-resolution ^1^H NMR analysis. Spectra were recorded in hexadeutero dimethyl sulfoxide (DMSO-d*6*; 600 μL for sample) purchased from Sigma-Aldrich (Milan, Italy) at 298 K, on a Bruker Avance Ultrashielded 300 MHz spectrometer equipped with a 5 mm multinuclear Z-axis gradient inverse probe head and at a proton frequency of 300.08 MHz. Relaxation times T1, pulse sequences, and acquisition and elaboration parameters were applied according to the literature [60,61,62].

### 3.3. Sample Preparation

A portion (2 g) of *Crocus sativus* L. stigmas was extracted, adopting the procedure already reported [63,64]. The enriched fraction was obtained by solid phase extraction (C18, 55 um, 70 A, Phenomenex, Torrance, CA, USA) as previously reported [65]. The fraction was freeze-dried in a vacuum centrifuge (Speed-Vac, Cryo Rivoire, Montpellier, France) and kept at 20 °C until its utilization.

### 3.4. Enzymatic Activity of the Purified Human Catalytic Fragment (cf-HMGR)

The enzymatic activity of *cf-HMGR* was spectrophotometrically determined as previously described [48]. Briefly, 0.85 μg of purified protein was added to assay buffer to reach a final volume of 100 μL, and this complete assay mixture (also containing 0.2 mM NADPH) was incubated at 37 °C. The reaction was started by adding 0.1 mM HMG-CoA to the complete assay mixture. The substrate-dependent oxidation of NADPH was measured at 340 nm using a UV spectrophotometer (Applied Biosystems model Jenway 7315, Thermo Fisher Scientific, Milan, Italy) equipped with a peltier unit. The rate of NADPH oxidation in the absence of HMG-CoA (control reaction) was subtracted from that obtained in the presence of HMG-CoA. NADPH oxidation was monitored every second for 10 min. Inhibition assays were performed by incubating the enzyme with different concentrations of picrocrocin from saffron extract (100–300 µg/mL) or pravastatin (100–300 nM).

### 3.5. Cell Cultures

The HepG2 cell line was purchased from the American Type Culture Collection (ATCC: Rockville, MD, USA) and cultured for maintenance purpose in DMEM-High Glucose (Sigma, St. Louis, MO, USA) and supplemented with 10% fetal bovine serum (FBS, Sigma), 1% penicillin/streptomycin (Sigma) and 2 mM L-glutamine (Sigma). Treatments were performed in the aforementioned medium without supplemented serum. Cells were cultured at 37 °C in 5% CO_2_ in a humidified atmosphere.

### 3.6. Cell Viability Assay

Cell viability was determined as previously described [66,67] using a 3-(4,5-Dimethyl-2-thiazolyl)-2,5-diphenyl-2H-tetrazolium bromide (MTT) assay. Briefly, cells were seeded in 48-well plates with a density of 2 × 10^4^ cells/well, and cultured in complete medium overnight. Cells were then treated with different concentrations of compounds for 24, 48 and 72 h. After treatment, MTT solution was added to each well (to a final concentration of 0.5 mg/mL) and plates were incubated at 37 °C for 2 h until the formation of formazan crystals. DMSO-solubilized formazan in each well was quantified by reading the absorbance at 570 nm using a microplate reader.

### 3.7. Quantitative PCR with Reverse Transcription (qRT–PCR)

Cells were grown in 10 cm dishes to reach 70–80% confluence, and exposed for 24/48 h to the vehicle (DMSO) or picrocrocin (from 1 to 300 µg/mL). Total cellular RNA was extracted using TRIZOL reagent (Invitrogen), as suggested by the manufacturer. RNA purity and integrity were assayed spectroscopically as well as by gel electrophoresis before analysis was performed. Complementary DNA (cDNA) was synthesized by reverse transcription, as already described [68]. For quantitative PCR (qPCR), primers based on the cDNA sequences of investigated genes were designed with Primer Express 3.0 (Applied Biosystems, Life Technologies, Carlsbad, CA, USA) and purchased from Invitrogen (Life Technologies) (Table 1). The qPCR reactions were performed using a Quant Studio7 Flex Real-Time PCR System (Life Technologies). An aliquot of 10 μL of reaction volume contained 25 ng of template (reverse transcribed first-strand cDNA), 5 μL SYBR Green Universal PCR Master Mix (BioRad, Hercules, CA, USA), and 300 nM of each specific primer [69,70]. The specificity of the PCR amplification was tested with the heat dissociation protocol following the final cycle of PCR. Each experiment was repeated at least 3 times [71,72]. The comparative threshold cycle method was used in relative gene quantification, as previously described [73,74], using Peptidylprolyl isomerase A (*PPIA*).

### 3.8. Immunoblotting Analysis

For immunoblotting analysis of the different proteins assessed in this study, cells were grown to 70–80% confluence and subjected to treatment. For total lysates’ preparation, cells were harvested and lysed in 200 µL of lysis buffer (50 mM Tris–HCl, 150 mM NaCl, 1% NP-40, 0.5% sodium deoxycholate, 2 mM sodium fluoride, 2 mM EDTA, 0.1% SDS) containing a mixture of protease inhibitors (aprotinin, phenylmethylsulfonyl fluoride, and sodium orthovanadate; Sigma). The same amounts of proteins from the total lysate or cytosolic fraction were resolved on SDS-polyacrylamide gel, transferred to a nitrocellulose membrane and probed with appropriate primary antibodies (Anti-SREBP1, Anti-SREBP2, Anti-GPAT, Anti-FASN, Anti-HMGR, Anti-LDLR, Santa Cruz Biotechnology, Santa Cruz, CA, USA and Merck KGaA, Darmstadt, Germany). To confirm equal loading and transfer, membranes were stripped and incubated with anti-GAPDH antibody [75] (Santa Cruz Biotechnology). The antigen–antibody complex was detected by incubating membranes with peroxidase-coupled goat anti-mouse antibody (Santa Cruz Biotechnology) and revealed using an ECL System (Bio-Rad Laboratories, Hercules, CA, USA) [76,77]. Blots were then exposed to film, and the bands of interest were quantified using ImageJ software (version 1.52a) [78].

### 3.9. Statistical Analysis

Data are presented as mean values ± standard deviation, taken over 3 independent experiments, with 3 replicates per experiment, unless otherwise stated. Statistical significance was measured using analysis of variance (ANOVA) test. A *p* value ≤ 0.05 was considered statistically significant [79].

## 4. Conclusions

The basic idea of this work was to obtain an extract from saffron, an easily available natural food enriched in organic compounds with healthy biological activities, without resorting to chemical-physical manipulations. This has allowed to obtain an extract enriched in picrocrocin (at least 90%) that is potentially safe for human food use.

For the first time, in this research work, the hypolipidemic properties of picrocrocin extracted from saffron stigmas have been highlighted. Interestingly, a non-statin-like mode of action seems to underlie the effects of this natural compound on lipid homeostasis. Remarkably, the observed increase in the *LDLR* expression level suggests a possible increase in LDL re-uptake with consequent lowering of blood cholesterol levels, thus reducing the cardiovascular risk associated with LDL cholesterol level. This healthy effect could be particularly beneficial in patients for whom statin therapy is contraindicated or who experience side effects after statin use. Furthermore, since an observed effect was the concurrent reduction of *FASN* and *GPAT* expression levels, we are confident that picrocrocin may reduce TG synthesis, thereby overcoming the known residual cardiovascular risk observed in patients on statin therapy. In addition, data from clinical studies showed that TG-rich lipoproteins and their cholesterol-enriched remnant particles have been related to atherogenesis [80]. Hence, the combined hypolipidemic effect obtained by administering an extract of natural origin could be an additional advantage.

## Figures and Tables

**Figure 1 ijms-24-03060-f001:**
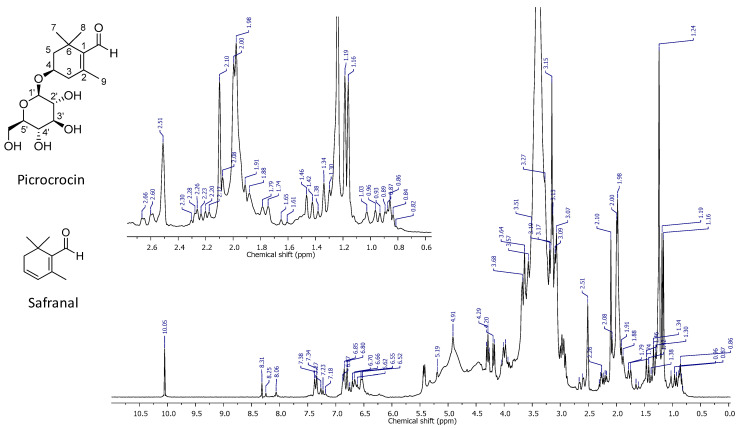
High-resolution ^1^H NMR spectrum of PEF.

**Figure 2 ijms-24-03060-f002:**
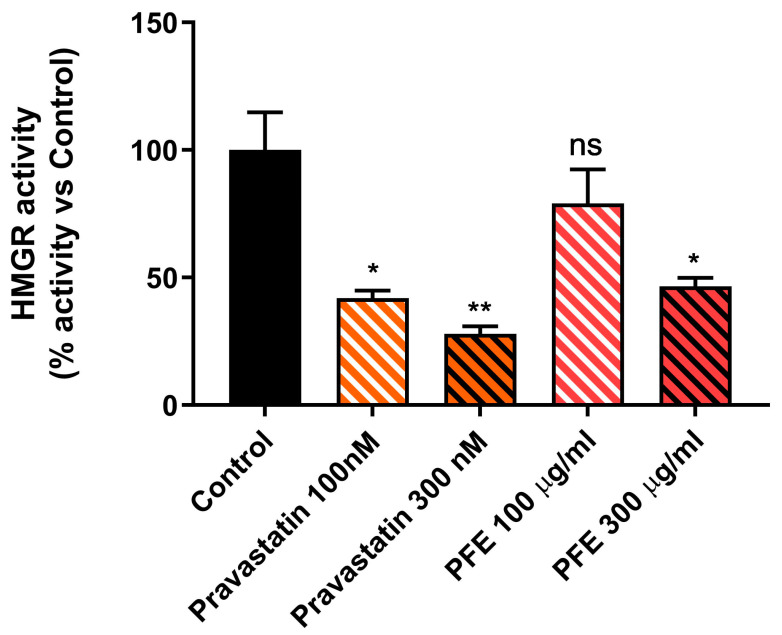
Picrocrocin inhibits the activity of the purified *HMGR* catalytic subunit. Results are expressed as percentage of enzymatic activity versus control (enzymatic assay without inhibitors); pravastatin was used as a positive control. Values represent mean ± SD of three independent experiments. * *p* value < 0.05; ** *p* value < 0.01; ns: non-significant.

**Figure 3 ijms-24-03060-f003:**
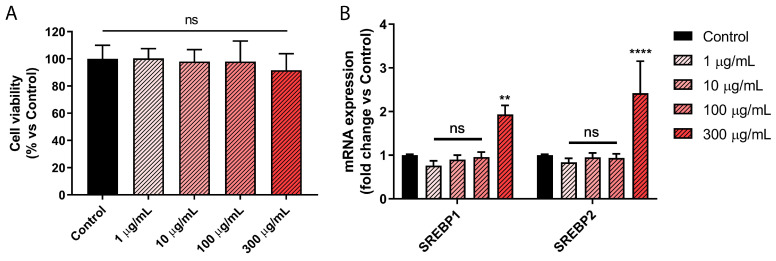
Effects of picrocrocin in HepG2 cells. (**A**) Cell viability assessment of HepG2 cells after treatment for 72 h, with increasing concentrations (1 to 300 µg/mL) of PEF from saffron extract. (**B**) qPCR analysis of *SREBP1* and *SREBP2* transcription levels after treatment for 24 h. The relative expression of human *SREBP1* and *SREBP2* were determined by Sybr green qPCR. The ∆Ct of human PPIA was used as an internal calibrator. These results highlight PEF’s ability to increase mRNA levels of *SREBP1* and *SREBP2* in HepG2 cells without affecting cell viability. Values represent mean ± SD of three independent experiments. ** *p* value < 0.01; **** *p* value < 0.0001; ns: non-significant.

**Figure 4 ijms-24-03060-f004:**
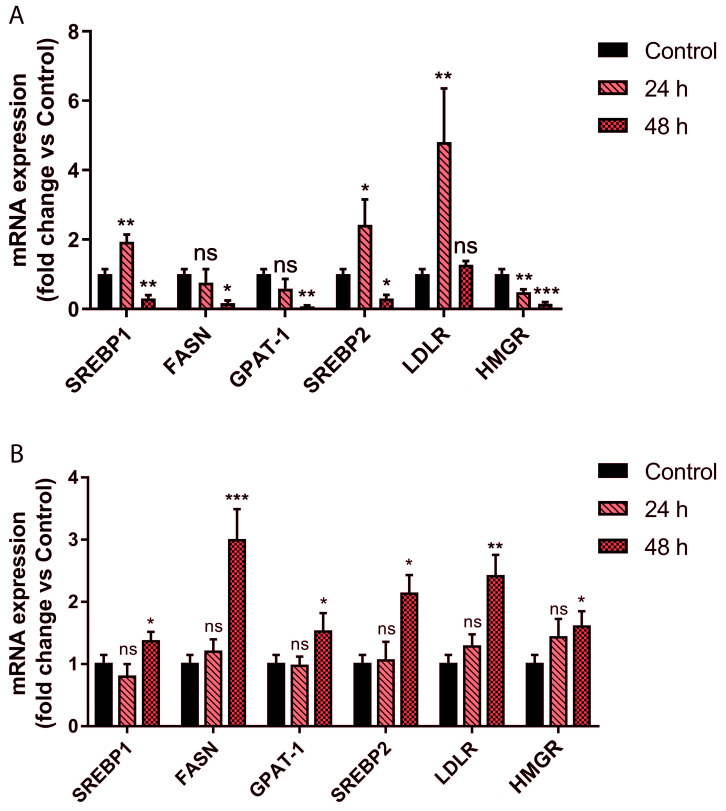
Effects of picrocrocin on the transcription of genes related to lipid metabolism. (**A**) qPCR analysis of the mRNA levels for the main proteins involved in lipid homeostasis after treatment of HepG2 for 24 and 48 h with PEF from our saffron extract (300 µg/mL). (**B**) qPCR analysis of mRNA levels after treatment of HepG2 cells for 24 and 48 h with pravastatin (5 µM). These results highlight PEF’s ability to modulate the mRNA levels of proteins involved in lipogenesis with a method different from that of pravastatin. Values represent mean ± SD of three independent experiments. * *p* value < 0.05; ** *p* value < 0.01; *** *p* value < 0.001; ns: non-significant.

**Figure 5 ijms-24-03060-f005:**
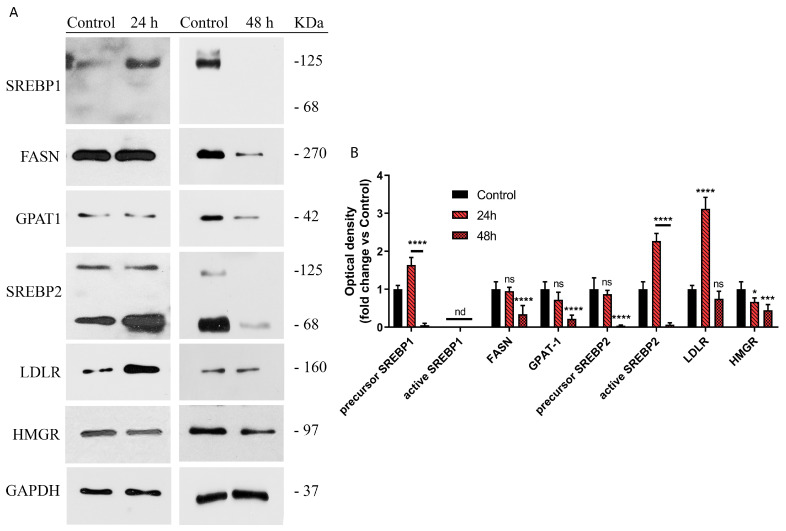
Effects of picrocrocin on protein expression of enzymes related to lipid metabolism. (**A**) Immunoblot analysis of the main proteins involved in lipid homeostasis after treatment of HepG2 cells for 24 and 48 h with PEF from our saffron extract. (**B**) Quantification of protein expression levels by densitometry. Values represent mean ± SD of three independent experiments. * *p* value < 0.05; *** *p* value < 0.001; **** *p* value < 0.0001; ns: non-significant; nd: not detectable.

**Table 1 ijms-24-03060-t001:** qPCR primers sequences.

Primer Name	Sequence (5′-3′)
SREBP1-Fw	GCGGAGCCATGGATTGCAC
SREBP1-Rv	TCTTCCTTGATACCAGGCCC
SREBP2-Fw	TGGCTTCTCTCCCTACTCCA
SREBP2-Rv	GCAGCTGCAAAATCTCCTCT
FASN-Fw	AGCTGCCAGAGTCGGAGAAC
FASN-Rv	TGTAGCCCACGAGTGTCTCG
GPAT1-Fw	GGCATCCTGAACTGGTGTGTG
GPAT1-Rv	GAGCTTGAGGAAGAGGATGGTG
HMGR-Fw	AGGTTCCAATGGCAACAACAGAAG
HMGR-Rv	ATGCTCCTTGAACACCTAGCATCT
LDLR-Fw	CAATGTCTCACCAAGCTCTG
LDLR-Rv	TCTGTCTCGAGGGGTAGCTG
PPIA-Fw	CATACGGGTCCTGGCATCTT
PPIA-Rv	TCCATGGCCTCCACAATATTC

## Data Availability

The data presented in this study are available on request from the corresponding authors.

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
