# Peer review of "A Picrocrocin-Enriched Fraction from a Saffron Extract Affects Lipid Homeostasis in HepG2 Cells through a Non-Statin-like Mode"

_ijms, 2023, doi:10.3390/ijms24043060_

Round 1

Reviewer 1 Report

Dear Authors,

There are several in vitro systems used to study drug metabolism and effects on the expression of metabolizing enzymes and the cellular response. In practice, the following are used: human liver microsomes (HLM), fraction and the S9 fraction of human liver, as well as hepatocytes, mainly primary human hepatocytes. However, each of these systems has its drawbacks. The best results are achieved using primary hepatocytes, which fully allow the biotransformation pathways to be recreated, to which the drug is subjected in the patient's body. However, short lifespan and genetic instability primary hepatocytes (decreased expression of CYP enzymes and other metabolizing proteins drugs) make it impossible to conduct research in a wider scope. In turn, in isolated liver cells or microsomes do not always express all enzymes metabolism or their level is too low to effectively study their role. Thus, the use of stable cancer cell lines has become commonplace. The choice of HepG2 cell line as the experimental model is therefore correct.

The manuscript submitted to me for evaluation presents a study interesting and significant both for its molecular and clinical aspects. In general, title and design are appropriate, accompanied by the methodology correctly chosen, accurately described and applied. Reservations may be raised only by the fact that over 60% of cited literature comes from more than 5 years. I pointed out typographical errors in the pdf version of the article.

Due to the increasing use of herbal medicine by the general community and the increased interest of some physicians, it is necessary to encourage research to confirm its biological effects based on modern scientific techniques. In this respect, the article deals with an important and current problem. A holistic approach to treatment involves combining modern medicine with other fields of science, including biochemistry, or pharmacology. Interdisciplinary cooperation works well in everyday clinical practice, bringing excellent results. The research carried out for the purposes of this publication can help in the therapy of potential patients who will benefit from medical intervention related to lipid metabolism disorders.

Author Response

Dear Authors,

Reviewer There are several in vitro systems used to study drug metabolism and effects on the expression of metabolizing enzymes and the cellular response. In practice, the following are used: human liver microsomes (HLM), fraction and the S9 fraction of human liver, as well as hepatocytes, mainly primary human hepatocytes. However, each of these systems has its drawbacks. The best results are achieved using primary hepatocytes, which fully allow the biotransformation pathways to be recreated, to which the drug is subjected in the patient's body. However, short lifespan and genetic instability primary hepatocytes (decreased expression of CYP enzymes and other metabolizing proteins drugs) make it impossible to conduct research in a wider scope. In turn, in isolated liver cells or microsomes do not always express all enzymes metabolism or their level is too low to effectively study their role. Thus, the use of stable cancer cell lines has become commonplace. The choice of HepG2 cell line as the experimental model is therefore correct.

The manuscript submitted to me for evaluation presents a study interesting and significant both for its molecular and clinical aspects. In general, title and design are appropriate, accompanied by the methodology correctly chosen, accurately described and applied. Reservations may be raised only by the fact that over 60% of cited literature comes from more than 5 years. I pointed out typographical errors in the pdf version of the article.

Authors response: We thank the reviewer for his/her comments; indicated typographical errors have been corrected accordingly. Some of the references have been updated to recent years; older references have been left only as needed.

Due to the increasing use of herbal medicine by the general community and the increased interest of some physicians, it is necessary to encourage research to confirm its biological effects based on modern scientific techniques. In this respect, the article deals with an important and current problem. A holistic approach to treatment involves combining modern medicine with other fields of science, including biochemistry, or pharmacology. Interdisciplinary cooperation works well in everyday clinical practice, bringing excellent results. The research carried out for the purposes of this publication can help in the therapy of potential patients who will benefit from medical intervention related to lipid metabolism disorders.

Reviewer 2 Report

In this study the authors explored the impact of a picrocrocin-enriched saffron extract on different aspects of the lipid metabolism of HepG2 hepatocytes. Saffron extracts have previously been reported to exhibit interesting biological properties but their effects on lipid metabolism have never been studied in detail. Here the authors present a high resolution 1H-NMR data of a picrocrocin-enriched saffron extract. However, these data are difficult to interpret but they simply indicate that this extract constitutes a highly complex mixture of different natural compounds, which includes picrocrocin but in addition a large number of other compounds. Next, the authors explored the impact of this highly heterogenous extract on different lipid metabolic aspects of HepG2 cells and found that high concentrations of these extracts slightly inhibited HMG-CoA reductase activity. At such high concentrations the expression levels of SREBP1, SRBPP2 and LDL-receptor was upregulated, whereas expression of HMG-CoA reductase was down. Although statistically significant the observed fold-change effects were rather subtle (2-3-fold increase). Similarly minor effects were observed when the steady state protein concentrations were quantified by immunoblotting. Unfortunately, there are a number of inconsistencies if one compares the PCR data with the outcome of the immunoblots. For instance, relative expression of the HMG-R protein (24 h) was not altered (Fig. 5) but its mRNA was significantly down (Fig. 4). Such opposing effects are difficult to explain. Why are the blotting gels of HMG-R after 24h and 48 h so different? There must be something wrong with the 48 h incubation. There is a similar problem with the the SREBP2 blots

            The major problem I have with this ms is that the authors employed a very heterogenous extract, but related the observed biological effect to one of these constituents (picrocrocin). The authors even do not know whether picrocrocin is one of the major constituents in the extract and the provided NMR data do not really answer this question. It would have been helpful if the authors could provide quantitative LC-MS data to analyze the chemical composition of their extracts and to quantify the picrocrocin content. If the authors are interested in the biological effects of crocin, picrocrocin and safranal why they did not perform corresponding biological experiments with the pure compounds? Their chemical structures are known and chemical synthesis should not be terribly difficult. Eventually, these compounds are even commercially available.

In the light of these comments, I feel myself unable to recommend this paper for publication in any scientific journal. To make the paper publishable more quantitative analytical data on the composition of the saffron extracts should be provided and experiments on the biological effects of pure extracts constituents should be carried out. In the absence of such additional data the value of the presented results is rather limited.

Author Response

Reviewer

 In this study the authors explored the impact of a picrocrocin-enriched saffron extract on different aspects of the lipid metabolism of HepG2 hepatocytes. Saffron extracts have previously been reported to exhibit interesting biological properties but their effects on lipid metabolism have never been studied in detail. Here the authors present a high resolution 1H-NMR data of a picrocrocin-enriched saffron extract. However, these data are difficult to interpret but they simply indicate that this extract constitutes a highly complex mixture of different natural compounds, which includes picrocrocin but in addition a large number of other compounds. Next, the authors explored the impact of this highly heterogenous extract on different lipid metabolic aspects of HepG2 cells and found that high concentrations of these extracts slightly inhibited HMG-CoA reductase activity. At such high concentrations the expression levels of SREBP1, SRBPP2 and LDL-receptor was upregulated, whereas expression of HMG-CoA reductase was down. Although statistically significant the observed fold-change effects were rather subtle (2-3-fold increase). Similarly minor effects were observed when the steady state protein concentrations were quantified by immunoblotting. Unfortunately, there are a number of inconsistencies if one compares the PCR data with the outcome of the immunoblots. For instance, relative expression of the HMG-R protein (24 h) was not altered (Fig. 5) but its mRNA was significantly down (Fig. 4). Such opposing effects are difficult to explain. Why are the blotting gels of HMG-R after 24h and 48 h so different? There must be something wrong with the 48 h incubation. There is a similar problem with the the SREBP2 blots

Authors response : We thank the reviewer for his/her comments. We apologize but actually, we made a mistake in preparing Figure 5 in the manuscript. Indeed, the original immunoblot of HMGR (24 hours) uploaded by the authors during the submission process was different, and it clearly displayed a slight reduction in HMGR expression after PEF treatment, which is consistent with the qPCR data reported in the manuscript. We have replaced figure 5 with the correct one. 

In the case of SREBP2 (after 24 h incubation) qPCR analysis detected an increase followed by a dramatic decrease (after 48 h incubation) in SREBP2 expression (FIG. 4A); it is well known, from consolidated literature data, that the SREBP-2 protein exerts sterol regulation through cleavage of the membrane-bound precursor protein to release the active form into the nucleus. In this specific regard, unlike Western blot analysis, qPCR analysis cannot discriminate between SREBP2 precursor and its cleaved mature form. Than according to the qPCR data, the related immunoblot highlighted a marked band due to proteolytic activation of the SREBP2 precursor (24h) that dramatically decreased at 48h (see FiG 5A), therefore the observed increase in SREBP2 mRNA level (24h) followed by its decrease (48h) is easily explained by the level of the SREBP-2 protein mature form.

            The major problem I have with this ms is that the authors employed a very heterogenous extract, but related the observed biological effect to one of these constituents (picrocrocin). The authors even do not know whether picrocrocin is one of the major constituents in the extract and the provided NMR data do not really answer this question. It would have been helpful if the authors could provide quantitative LC-MS data to analyze the chemical composition of their extracts and to quantify the picrocrocin content. If the authors are interested in the biological effects of crocin, picrocrocin and safranal why they did not perform corresponding biological experiments with the pure compounds? Their chemical structures are known and chemical synthesis should not be terribly difficult. Eventually, these compounds are even commercially available.

In the light of these comments, I feel myself unable to recommend this paper for publication in any scientific journal. To make the paper publishable more quantitative analytical data on the composition of the saffron extracts should be provided and experiments on the biological effects of pure extracts constituents should be carried out. In the absence of such additional data the value of the presented results is rather limited.

Authors: We thank the reviewer for his/her comments, which offer the possibility to better explain the aim of this work. The basic idea of the work was to obtain an extract from readily available food and natural matrices, enriched in organic compounds characterized by well-known and remarkable biological activities, and subject it, without chemical and physical manipulations, to further targeted and specific biological tests in order to check it for further biological potential.

In order to prepare this type of extract, a well-known extraction methodology has been applied to the commercially available matrix, a rapid and straightforward methodology that is widely accepted by the scientific community, and also useful in the case of other different complex plant matrices (see, also: Foods, 2022, 11, 3245; Journal of Applied Botany and Food Quality 2022, 95, 105-113; Trends in Food Science and Technology 2019, 89, 26-44).

In our case, the obtained extract showed a quali-quantitative NMR profile (with the relative quantification based on the value of the intensities of the resonance peaks selected for the attribution of the known structures of the compounds of interest) which the plot contour and values of the signal integrals close to that illustrated in the article mentioned above, indicating a picrocrocin content of not less than 90%.

On the basis of these results, and also based on the comparison of our experimental data with those reported in the literature, the Authors estimated the picrocrocin content to be not less than 90% also in the extract used as the case study, consequently they excluded the presence of crocins that could have interfered in the evaluation of the bioactivity of the extract.

It was therefore not considered appropriate to resort to further instrumental analyses, which, unlike NMR, can introduce artifacts in quantifications, and which also require the use of the same organic compounds to be quantified in their deuterated form, as the standards for the preparation of spiked matrices.

Furthermore, the compounds present in higher concentration in the extract are difficult to find in the form of an analytical purity grade, an absolutely necessary characteristic when commercial organic compounds must be used as reference compounds for quantitative purposes.

Finally, the work presented in the manuscript was not carried out by a synthetic organic chemistry laboratory, and it should not be forgotten that the use of synthetic standards must be validated before their application as reference compounds in order to estimate the real content of different targeted metabolites in complex natural matrices.

Reviewer 3 Report

The paper submitted for review is a significant contribution to the development of pharmacology and pharmacotherapy. Unfortunately, the manuscript has some minor errors and omissions.

1.      How authors choose concentrations of drugs used in experiment?

2.      The text requires editorial corrections, e.g. ND: undetectable in lowercase, some consequences should be introduced in the figure captions.

Author Response

Reviewer

The paper submitted for review is a significant contribution to the development of pharmacology and pharmacotherapy. Unfortunately, the manuscript has some minor errors and omissions.

How authors choose concentrations of drugs used in experiment?

Authors response:

Pravastatin concentration used for HMGR-cf activity assay was chosen according to our previous work (Curcio R. et al. 2020, Molecular Biotechnology, 62(2), 119-131). For HepG2 treatment, pravastatin concentration was chosen according to previous works (Cohen L.H. et al 1993, Biochemical Pharmacology; Kawata S. et al 1994, British journal of cancer; Bartolomei M. et al 2022, Nutrients), which highlighted a good inhibition of cholesterol synthesis by treating HepG2 cells with 1 to 10 µM Pravastatin.

Reviewer

The text requires editorial corrections, e.g. ND: undetectable in lowercase, some consequences should be introduced in the figure captions.

Authors response: According to the suggestions of the reviewer, some consequences have been introduced in the figure captions, as well as some corrections have been made in the text and figures.

Round 2

Reviewer 2 Report

In their rebuttal letter the authors replied to the critical comments I made during the first round of evaluations of this ms. Their argumentation is straight forward and although I do not agree with all what the authors saying I accept most of their arguments. Unfortunately, hardly any of these arguments have been considered for preparation of the revised version of the ms. In fact, accept of minor changes the ms was hardly modified. For me, this may not be acceptable. In their rebuttal letter the authors stated that my criticisms gave them the opportunity to explain the aim of their studies in more detail. Why was this paragraph not included into the ms. When I read the original version of the ms I did not really understand why the authors did not carry out the experiments with purified compounds instead of the natural extracts. The revised ms does not really refer to this question. Since other readers may also have this question, the authors should explain in the ms why they used this experimental approach and not an alternative strategy which others including myself may have preferred.

In general, state of the art revision of the scientific ms requires discussion of the critical points raised by the reviewers. To do so only in the rebuttal letter may not be sufficient. Thus, in my opinion the ms still needs major textual revision.

Author Response

In their rebuttal letter the authors replied to the critical comments I made during the first round of evaluations of this ms. Their argumentation is straight forward and although I do not agree with all what the authors saying I accept most of their arguments. Unfortunately, hardly any of these arguments have been considered for preparation of the revised version of the ms. In fact, accept of minor changes the ms was hardly modified. For me, this may not be acceptable. In their rebuttal letter the authors stated that my criticisms gave them the opportunity to explain the aim of their studies in more detail. Why was this paragraph not included into the ms. When I read the original version of the ms I did not really understand why the authors did not carry out the experiments with purified compounds instead of the natural extracts. The revised ms does not really refer to this question. Since other readers may also have this question, the authors should explain in the ms why they used this experimental approach and not an alternative strategy which others including myself may have preferred.

In general, state of the art revision of the scientific ms requires discussion of the critical points raised by the reviewers. To do so only in the rebuttal letter may not be sufficient. Thus, in my opinion the ms still needs major textual revision.

Authors: Following the suggestions of reviewer we have modified the revised version of our manuscript in Sections 2.1, 2.2, 2.3, 3.2 and in the Conclusion section, including the arguments in response to all reviewers’ critical comments contained in the rebuttal letters. We hope this helps readers better understand our work. All changes have been tracked.

Round 3

Reviewer 2 Report

The authors modified the text according to my suggestions and now the ms can be published as it is.